# 3D genome of multiple myeloma reveals spatial genome disorganization associated with copy number variations

Pengze Wu[1], Tingting Li[1], Ruifeng Li[1], Lumeng Jia[1], Ping Zhu[1], Yifang Liu[2], Qing Chen[1], Daiwei Tang[2], Yuezhou Yu[1] & Cheng Li[1,3]

The Hi-C method is widely used to study the functional roles of the three-dimensional (3D) architecture of genomes. Here, we integrate Hi-C, whole-genome sequencing (WGS) and RNA-seq to study the 3D genome architecture of multiple myeloma (MM) and how it associates with genomic variation and gene expression. Our results show that Hi-C interaction matrices are biased by copy number variations (CNVs) and can be used to detect CNVs. Also, combining Hi-C and WGS data can improve the detection of translocations. We find that CNV breakpoints significantly overlap with topologically associating domain (TAD) boundaries. Compared to normal B cells, the numbers of TADs increases by 25% in MM, the average size of TADs is smaller, and about 20% of genomic regions switch their chromatin A/B compartment types. In summary, we report a 3D genome interaction map of aneuploid MM cells and reveal the relationship among CNVs, translocations, 3D genome reorganization, and gene expression regulation.

[1] Peking-Tsinghua Center for Life Sciences, Academy for Advanced Interdisciplinary Studies, Center for Bioinformatics, School of Life Sciences, Peking University, Beijing 100871, China. [2] PKU-Tsinghua-NIBS Graduate Program, School of Life Sciences, Tsinghua University, Beijing 100084, China. [3] Center for Statistical Science, Peking University, Beijing 100871, China. Pengze Wu, Tingting Li and Ruifeng Li contributed equally to this work. Correspondence and requests for materials should be addressed to C.L. (email: cheng_li@pku.edu.cn)

Aneuploid genomes with whole or partial chromosomal gains and losses are observed in more than 70% of cancers and non-cancerous diseases such as Down syndrome[1]. Aneuploidy can be caused by genome instability and mitotic defects, which leads to cellular responses such as cell cycle delay and slow growth[2]. Under certain stress conditions aneuploidy can also increase cell survival[3]. Various mechanisms are proposed for these aneuploidy phenotypes. Copy number variations due to aneuploidy affect mRNA and protein expression of cancer-related genes and downstream proliferation pathways[4]. Aneuploidy further induces chromosome mis-segregation and genome instability[5] and thus promotes tumor cell evolution via feedback loops[6]. Despite these findings, the role of aneuploidy in cancer initiation and progression is not fully understood[6].

Chromatin conformation capture techniques, such as Hi-C and ChIA-PET, have recently been developed to probe the three-dimensional (3D) genome organization of genomes at high resolution[7, 8] and reveal gene regulation mechanisms. Studies using these techniques have found that the mammalian genome is organized into gene-dense and transcriptionally active compartment A as well as gene-sparse and transcriptionally inactive compartment B at the megabase scale[7]. Topologically associating domains (TADs) are formed at the sub-megabase scale, which are function units for regulating gene expression and overlap with replication domains[9, 10]. Within TADs, chromatin loops facilitate long-range interactions between enhancers and promoters for gene regulation[11]. The 3D organization of the genome is dynamically regulated in key biological processes such as cell division[12], stem cell differentiation[13], and B-cell activation[14].

Chromosome conformation capture techniques have recently been applied to the study of cancer genomes[15]. Hi-C data of breast cancer genomes have revealed that the switching of compartments A/B between normal and cancer cells is associated with changes in gene expression[16]. TADs in prostate cancer cells are smaller than those in normal prostate cells and are altered at the TP53 tumor suppressor locus[17]. Conversely, the spatial organization of the genome also shapes cancer genome alterations[18]. For example, frequencies of translocation partner choices are associated with the probability of spatial contact of two involved loci[19]. Translocations are prone to occur at hot spots of double-strand breakpoints located in regions that have high spatial proximity to each other[20]. These studies suggest that the 3D organization of the genome and cancer genome alterations reciprocally influence each other.

Despite the progress entailed by these findings, few 3D cancer genome data sets have been established. Since cancer is frequently associated with genomic alterations such as aneuploidy, copy number variation (CNV), and mutation, it is important to characterize the spatial disorganization of the cancer genome and determine its functional consequences. In this study, with the aim of better understanding the molecular mechanism of aneuploidy cancers such as multiple myeloma (MM), we applied an integrated analysis combining Hi-C data, whole-genome sequencing (WGS) data, and RNA sequencing data on two aneuploid MM cell lines. We investigated the CNV-driven bias in Hi-C data of aneuploid cancers. We discovered association between CNVs, translocations and 3D genome architectures. We also reported changes in gene expression associated with the disorganized 3D genome, which are implicated in the development of MM.

## Results

**Correcting copy number variation bias in cancer Hi-C data.** MM is a cancer developed from antibody-generating plasma B cells and has two major subtypes, hyperdiploid MM and non-hyperdiploid MM[21]. We used two MM cell lines (RPMI-8226 and U266) to study the genome-wide chromatin interactions of

aneuploid cancers. We first confirmed the chromosome aneuploidy through karyotyping experiments. The RPMI-8226 genome is nearly triploid with multiple trisomy and tetrasomy chromosomes (Supplementary Fig. 1a, b), while the U266 genome is nearly diploid with only a few chromosomal gains or losses (Supplementary Fig. 1c, d). The karyotypes of the two MM cell lines were further confirmed by CNV analysis of WGS data sets (Supplementary Fig. 1e). Next, we performed in situ Hi-C experiments of these two cell lines with two restriction enzymes (HindIII and MboI), and sequenced approximately 200 million reads for each replicate of the two cell lines. After data analysis (see material and methods) we obtained chromatin interaction heatmaps at 40-kb resolution for each sample using combined data of replicates (Supplementary Table 1). Data processing showed high quality of the Hi-C data and reproducibility of the Hi-C replicates, demonstrating the successful performance of Hi-C experiments (Supplementary Fig. 2a–d).

The whole-genome interaction maps showed that about 70% of interactions occurred within chromosomes and 30% between chromosomes. This intra/inter-chromosome interaction ratio is similar to that of diploid cells as reported previously[11] (Fig. 1a and Supplementary Fig. 3a). We reasoned that trisomic chromosomes are sampled more frequently than disomic ones in a Hi-C library, so chromosome interactions involving chromosome regions with higher copy numbers would have higher interaction counts. By applying the same CNV calling method to WGS data and Hi-C data, we obtained similar CNV results from the two data types with high correlation coefficients (Fig. 1b, c and Supplementary Fig. 3b, c). We also found that the average interaction count inside each CNV block was positively correlated with its copy number (Fig. 1d and Supplementary Fig. 3d). These results indicate that raw interaction counts in cancer Hi-C data are biased by CNVs and should be corrected to obtain per-copy chromosome interaction map. We compared the ICE[22] and HiCNorm[23] methods for normalizing the interaction matrix. ICE can better correct the CNV bias than HiCNorm (Fig. 1e, f and Supplementary Fig. 3e, f), so we used it to obtain normalized Hi-C matrices for downstream analysis.

**3D genome is influenced by inter-chromosomal translocations.** Besides CNVs, we found that genomic structure alterations such as translocations can be reflected by Hi-C interaction matrices. We reasoned that high inter-chromosomal interactions in Hi-C matrices are likely due to chimeric chromosomes, which were caused by translocation events (black boxes in Fig. 1a and Supplementary Fig. 3a). To confirm this, we identified translocations from the WGS data by using the CREST software[24] and compared them with Hi-C inter-chromosomal interactions. In RPMI-8226, we identified 56 inter-chromosomal translocation events (Fig. 2a), four of which were also in the top 100 highest Hi-C inter-chromosomal interactions (blue links in Fig. 2a, b, Fisher's exact test p-value: $2.565 \times 10^{-8}$), and the observed/expected ratio is 90.91 (Supplementary Fig. 4a). To further check whether high inter-chromosomal interactions represent chimeric chromosomes, we referenced the spectral karyotyping (SKY) image of RPMI-8226 (http://www.ncbi.nlm.nih.gov/sky/). A chimeric chromosome was detected by SKY, involving chromosome 16 and 22 (Fig. 2c). This translocation of t(16, 22)(q23, q11) was identified by both WGS data and Hi-C data (Fig. 2d–f), which involves MM-related genes WWOX[25] and MAF[26]. By integrating the WGS data and Hi-C interactions of RPMI-8226, we identified additional inter-chromosomal translocations supported by both data types, involving cancer-related genes ADORA2B[27], FLII[28], AMBRA1[29] and PTRPJ[30] (Supplementary Fig. 4, Supplementary Data 1). In U266 cells, we identified 80 genes affected by the translocation events, including the cancer-related genes TNIK[31],

*FBXW7*[32] and *TRIM2*[33] (Supplementary Fig. 5). Some of these genes have not been implicated in MM before. We also investigated genes that locate within 1 Mb distance from the translocation sites, and the result showed that over a half of them were differentially expressed compared to the GM12878 B cells (Supplementary Data 2). The inter-chromosomal interactions

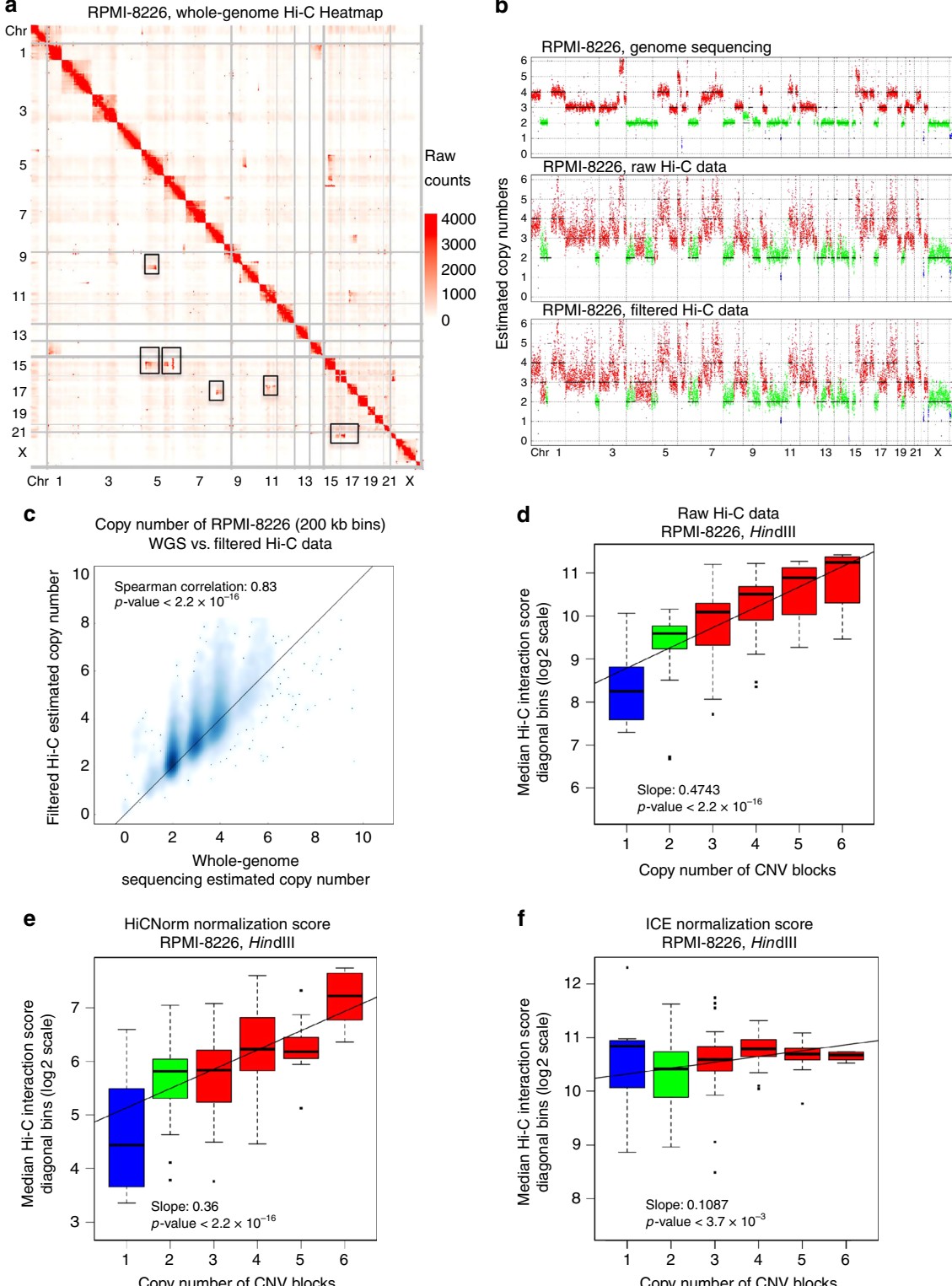

**Fig. 1** Correcting copy number variation in cancer Hi-C data. **a** Whole genome Hi-C interaction matrix of RPMI-8226 cells. Black rectangles indicate inter-chromosomal translocations. **b** CNVs of RPMI-8226 cells estimated from different sequencing data sets. Methods for filtered Hi-C data are described in the Material and Methods. **c** Scatterplot between the copy numbers estimated from WGS (*x*-axis) and filtered Hi-C data (*y*-axis) shows high similarity. **d** Median raw Hi-C interaction counts of the diagonal bins in each CNV block vs. the copy number of CNV blocks. A CNV block is defined as a continuous chromosome region with the same estimated copy number from WGS data. **e**, **f** Same as **d**, but the *y*-axis data are HiCNorm **e** and ICE-normalized **f** interaction values. The *p*-values of figures **d**–**f** were determined by F-test

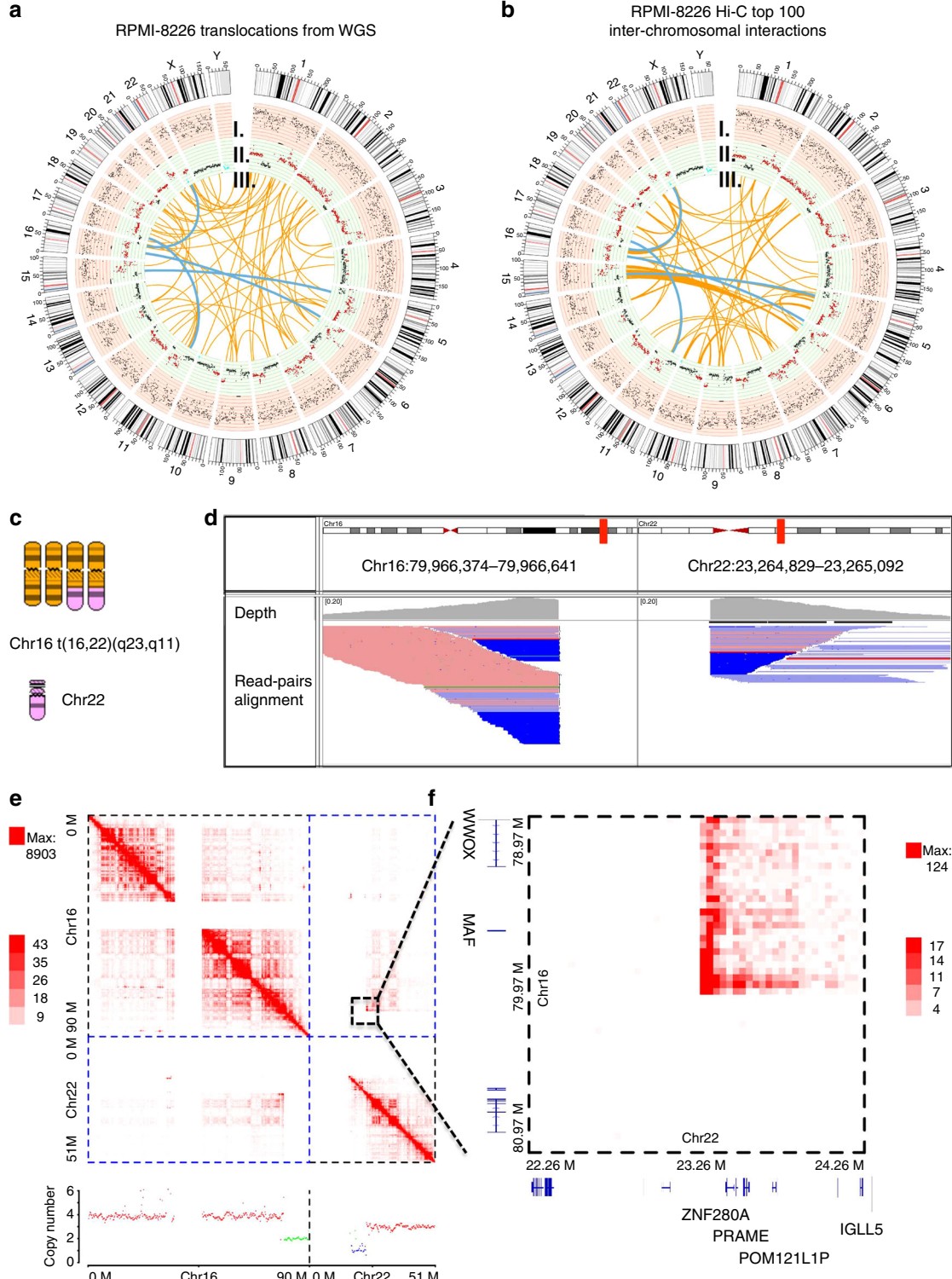

**Fig. 2** Hi-C data reveal translocation events. **a** Fifty-six inter-chromosomal translocation events identified in RPMI-8226 by WGS. From outer circle to inner circle: panel I: average RNA-seq count for every 200 kb chromosome bin; panel II: CNV data from WGS (black dots: copy number equal to 2, red dots: copy numbers larger than 2, green dots: copy numbers less than 2); panel III: orange lines: translocation events, blue lines: translocation events that are among the top 100 highest inter-chromosomal Hi-C interactions in **b**. **b** The 100 highest inter-chromosomal Hi-C interactions in RPMI-8226 cells (5 M resolution): from outer circle to inner circle: panel I and II: the same as in **a** for RNA-seq counts and CNV data; panel III: orange lines: top 100 highest interactions, blue lines: the common events with **a**. **c** The chimeric chromosome formed between chr16 and chr22 identified by spectral karyotyping (SKY). The SKY image of RPMI-8226 is from the NCBI SKY database. **d** The translocation event of t(16;22)(q23.2;q11.22) in RPMI-8226 cells plotted by IGV tool, showing only WGS paired reads supporting this translocation event. The reads depth and alignment panels are shown. Read alignments are sorted by read location and colored by read strands. **e** Top panel: the intra and inter-chromosome interaction maps of chromosome 16 and 22 in RPMI-8226. Bottom panel: copy number variations of chromosome 16 and 22. **f** Enlarged inter-chromosomal interactions corresponding to the translocation t(16;22)(q23.2;q11.22)

identified from the MM cell lines showed higher overall interaction counts than those from normal B cells (Supplementary Fig. 5f). Collectively, these results showed that the 3D cancer genome is influenced by inter-chromosomal translocations and cancer Hi-C data reflect chromatin interactions from both normal and chimeric chromosomes.

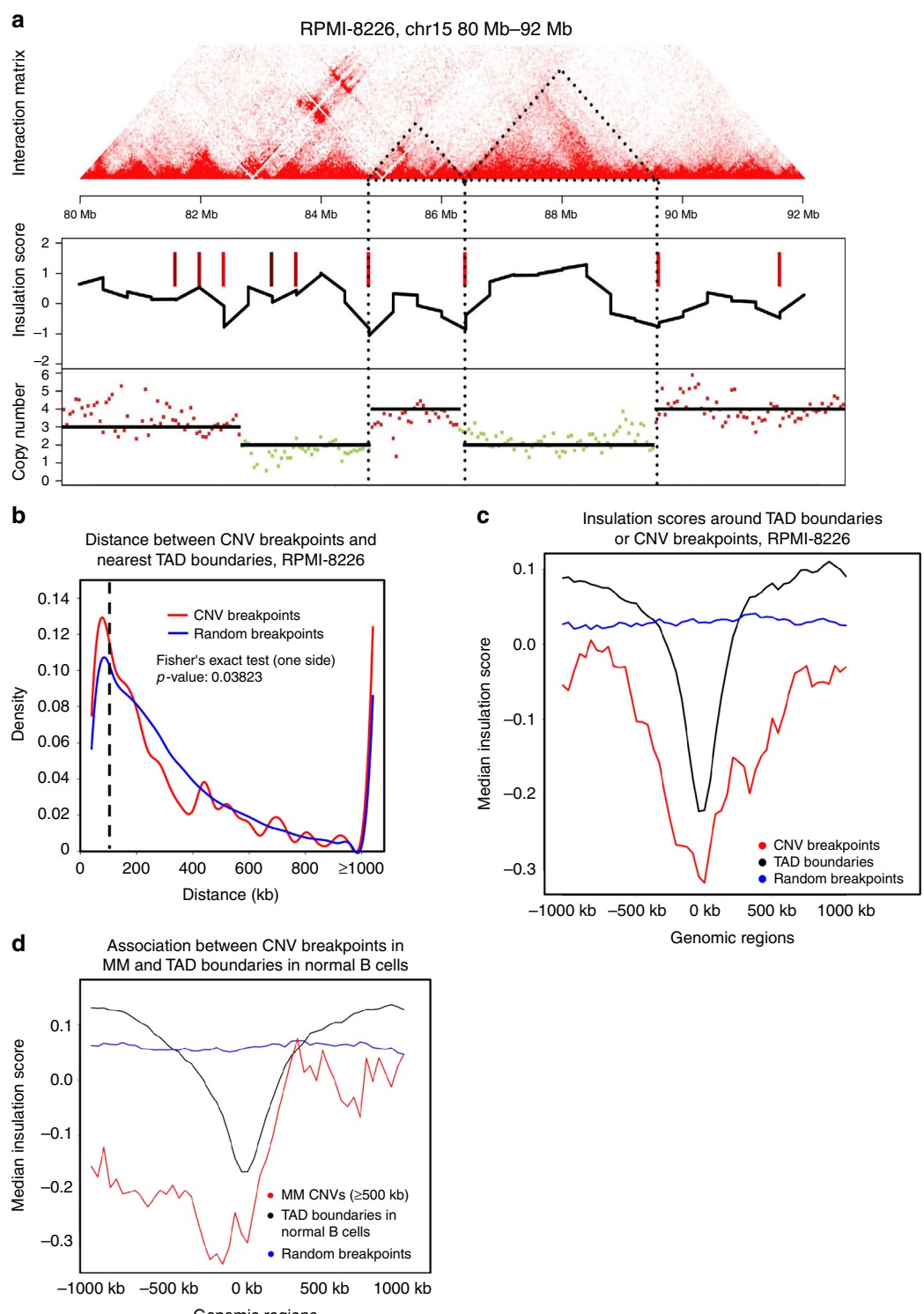

**Fig. 3** CNV breakpoints associated with TAD boundaries. **a** Hi-C interaction matrix of a region (chr15: 80 Mb-92 Mb) in RPMI-8226 shows the association between TAD boundaries and CNV breakpoints. Top: Hi-C interaction matrix, middle: TAD boundaries (vertical bars) and insulations scores, bottom: copy number variations. The dashed triangles indicate two TADs coinciding with CNV blocks. **b** The distribution plots of the distance between all CNV breakpoints (red line) or random sites (blue line) to their nearest TAD boundaries in RPMI-8226 cells. **c** The averaged insulation scores at TAD boundaries (black line) or CNV breakpoints (red line) in RPMI-8226 are lower compared to surrounding regions, in contrast to random sites (blue line). **d** The averaged insulation scores around different sets of genomic locations in B cells (GM12878). Blue line: randomly selected sites; black line: TAD boundaries in GM12878; red line: CNV breakpoints from the CNVD database

**CNV breakpoints are associated with TAD boundaries**. A recent study of the 3D prostate cancer genome found that CNVs may help establish new TAD boundaries[17]. We thus investigated the relationship between CNVs and TADs in MM cells. ICE-normalized interaction matrices were used to call TADs and WGS data were used to call CNV blocks, both at 40-kb resolution. In total, we identified 3457 TAD boundaries and 596 CNV breakpoints in RPMI-8226 cells. We found that CNV breakpoints often occur near TAD boundaries (Fig. 3a). A total of 7.5% of all CNV breakpoints were also TAD boundaries and 30.7% of the CNV breakpoints

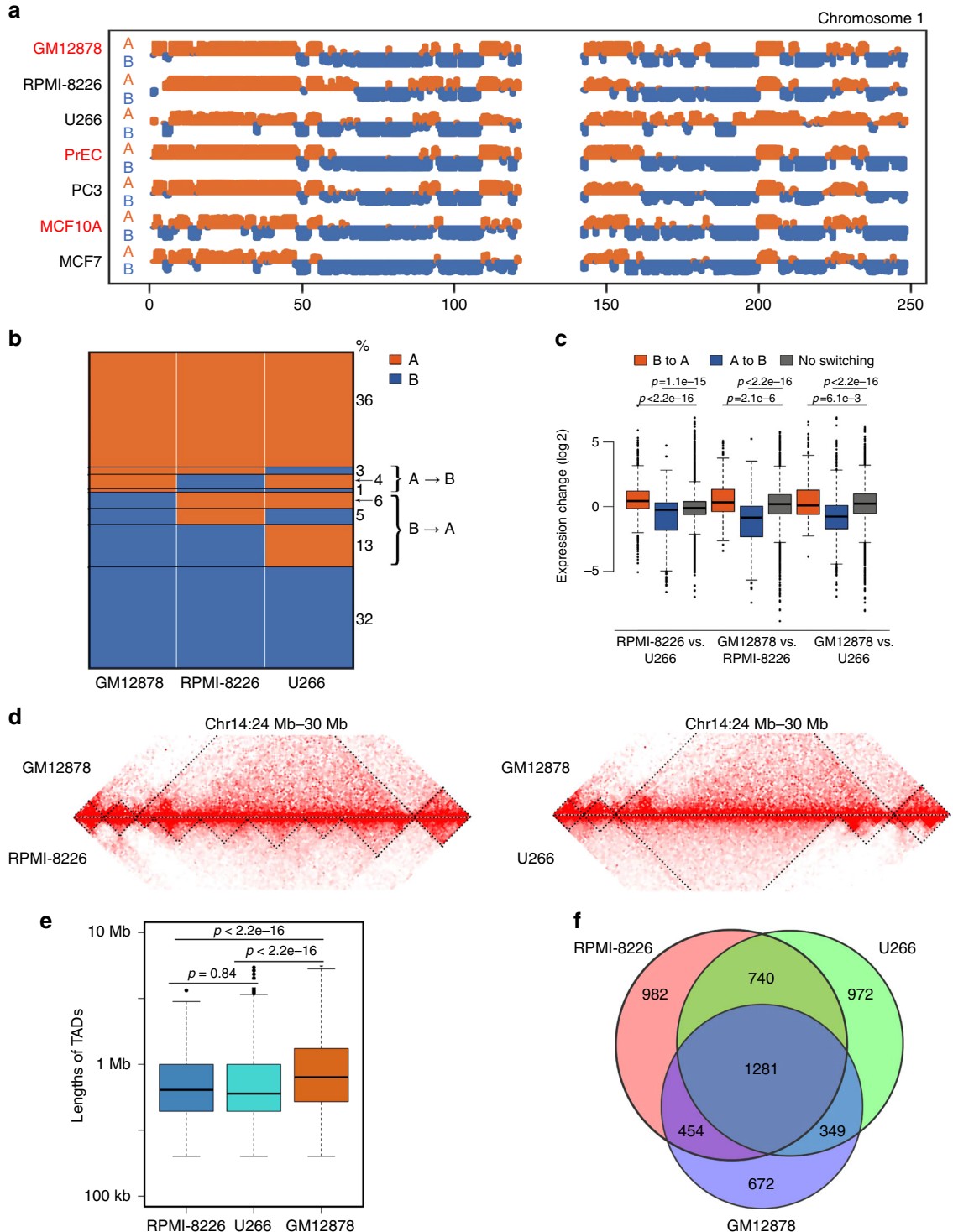

**Fig. 4** Reorganization of 3D chromosome is associated with gene expression changes. **a** The A/B compartments in RPMI-8226 and U266 compared with normal B cells (GM12878) in chromosome 1. **b** Genome-wide proportions of A/B compartment changes among RPMI-8226, U266 and GM12878 cells. **c** Boxplots of expression changes of genes grouped by their A/B compartment changes (t-test). **d** Example of conserved and changed TADs in a region (chr14: 24 Mb-30 Mb) comparing RPMI-8226, U266 and GM12878 cells. **e** The average size of TADs in multiple myeloma cells compared with GM12878. **f** The number of conserved and changed TADs in multiple myeloma cells compared with GM12878

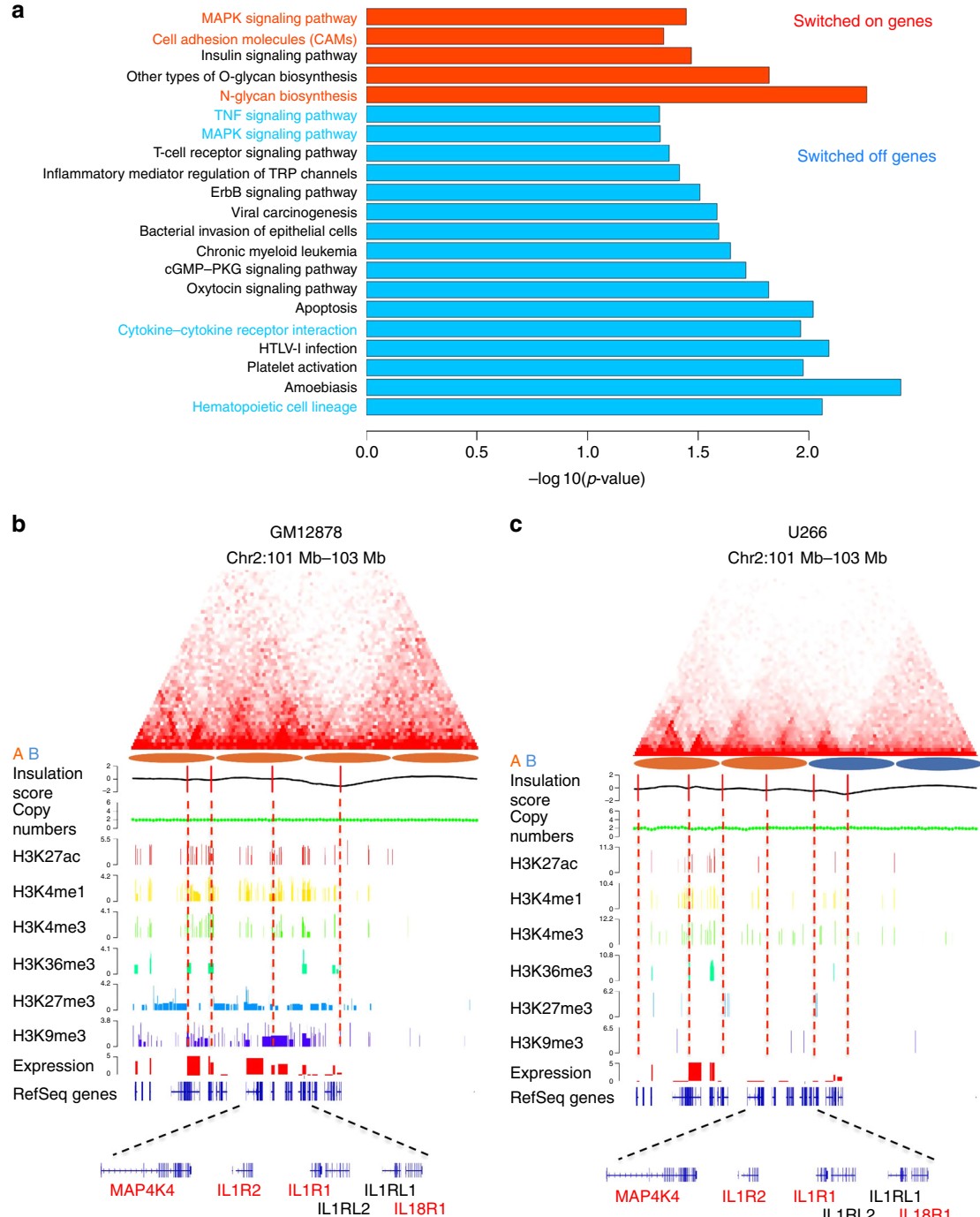

**Fig. 5** The 3D structure and expression changes of multiple myeloma-related genes. **a** Gene pathway enrichment analysis for the switched on/off genes. Switched on genes are defined when their genomic compartment changed from B to A in both MM cell lines (compared with GM12878). Switched off genes are defined when their genomic compartment changed from A to B in both MM cell lines (compared with GM12878). Genes with mutations that affect protein coding were discarded. **b** Example of cytokine receptor gene cluster including IL1R1, IL1R2, IL18R1. The *IL1R2* gene is located at chr2:101991844-102028544, and the plotted locus is from 1 Mb upstream to 1 Mb down-stream region surrounding the *IL1R2* gene. Red color gene names indicate the genes that are enriched in the cytokine-cytokine receptor interaction pathway. In GM12878, IL1R1, IL1R2, IL18R1 and MAP4K4 were highly expressed. Insulation score, TAD boundaries (vertical red lines), copy numbers, ChIP-seq peaks of H3K27ac, H3K4me1, H3K4me3, H3K36me3, H3K27me3, H3K9me3, RNA-seq expressions and RefSeq gene tracks are shown below the Hi-C heatmaps. H3K4me3 is annotated as promoters, H3K4me1/H3K27ac as active enhancers, H3K36me3 as transcribed gene body and H3K9me3 as heterochromatin. **c** The same genomic region as in **b** using the data of U266 cells. The expression of IL1R1, IL1R2, IL18R1 and MAP4K4 are downregulated and the TAD boundaries are altered, accompanied by reduced epigenetic marks such as H3K27ac, H3Kme1 and H3K4me3

located within 120 kb of TAD boundaries (Fig. 3b). The distances between CNV breakpoints and their nearest TAD boundaries were significantly smaller than location-randomized breakpoints.

Since insulation scores can be used to identify TAD boundaries[34], we averaged insulation scores within 1 Mb of all TAD boundaries or all CNV breakpoints. The average insulation scores at both TAD boundaries and CNV breakpoints were lower than that of surrounding regions (Fig. 3c). The results were similar in U266 cells (Supplementary Fig. 6a–c), confirming the association of CNV breakpoints and TAD boundaries in MM cells. This association can be explained by two possible causes: (i) CNVs can induce the formation of new TADs with boundaries close to CNV breakpoints, or (ii) CNV breakpoints occurred more frequently near TAD boundaries in normal cells, or that during cancer cell evolution the CNVs that disrupt TAD structures were preferentially lost. To check the second possibility, we correlated a database of cancer CNVs[35] and TAD boundaries in normal B cells (GM12878). We found that in normal B cells, the average insulation score at the myeloma CNV breakpoints was also lower than that in the surrounding regions and differed from that at random breakpoint sites (Fig. 3d), suggesting that TAD boundaries in normal cells are also associated with cancer CNV breakpoints. These results support the second assumption that either CNV breakpoints are prone to occur at TAD boundaries in normal cells or TADs are resistant to disruption during transformation from normal cells to cancer.

**Association between A/B switches and gene expression.** The mammalian genome consists of actively transcribed compartments A and inactive compartments B[7]. Switching of compartments A/B between normal cells and breast cancer cells is associated with corresponding changes of gene expression[16]. We investigated whether this phenomenon is exhibited in MM. We compared MM cells, which are of a B-cell lineage[21], with a lymphoblastoid B-cell line (GM12878). We determined the compartment types of the genome at 500-kb resolution in GM12878, U266, and RPMI-8226 cells using the HiTC package[36]. Most genomic regions remained in the same compartments in MM cells compared to normal B cells (Fig. 4a, b). A total of 8% of genomic regions switched from the compartment A in normal B cells to compartment B in one or both MM cells and associated with downregulated gene expression. 24% of genomic regions exhibited the opposite switching from compartment B in normal cells to compartment A in cancer cells and associated with up-regulated gene expression (Fig. 4b, c). These results are consistent with the previous study on breast cancer[16].

We next compared TADs in MM cells and B cells. We called TADs from Hi-C interaction matrices at 40-kb resolution[37] and identified 2756, 3457 and 3342 TADs in GM12878, RPMI-8226 and U266 cells, with their median TAD sizes of 800, 600 and 640 kb, respectively (Fig. 4d). The numbers of TADs increased by 1.2-fold in both MM cells compared to B cells, and the average size of TADs in MM was smaller (Fig. 4e). We defined two TADs as being conserved between two samples if they overlapped by more than 70% of both TAD regions. We identified 1281 TADs that were conserved among all the three samples and 740 TADs that were conserved in the two MM cells but not in normal B cells (Fig. 4f). This is consistent with recent findings showing an increased number of TADs in prostate cancers compared with that in normal prostate cells[17]. These results indicate that the 3D genome architecture of MM cells are reorganized and is associated with gene expression regulation.

**Chromatin conformation and gene expression changes.** We next asked whether reorganization of the 3D genome in MM has

functional consequences on MM pathology. To explore the functional effects of 3D genome changes between B cells and MM, we performed KEGG pathway analysis with genes located in the 6% of genomic regions that switched from compartment B in B cells to compartment A consistently in both MM cells, and genes located in the 1% of genome regions that consistently switched in the opposite direction in both MM cells (Fig. 4b). The pathway enrichment analysis showed that these genes are strongly associated with MM-related pathways, including the MAPK signaling pathway[38], N-glycan biosynthesis[39], TNF signaling pathway[40] and cytokine–cytokine receptor interaction pathway[41] (Fig. 5a). Within the consistent A/B compartment-switching regions, we found a cytokine receptor gene cluster at 2q11.2-q12.1[42] which had A-type compartment in normal B cell but B-type compartment in both MM cells, and were associated with downregulated gene expression involving several interleukins *IL1R1*, *IL1R2*, *IL18R1* and the cytokine *MAP4K4*. The gene expression changes of this locus are associated with altered TAD structures and epigenetic markers but not copy number variations (Fig. 5b, c). The TAD boundaries nearby the *IL1R2* gene are changed in U266 compared with GM12878, and the signals of active enhancer markers H3K27ac and H3K4me1 are reduced in U266 compared with GM12878. By combining Hi-C data, WGS and gene expression data, the spatial disorganization and gene expression changes in MM can be associated at the TAD and gene level. It should be noted that some changes between GM12878 and MM cells might be due to the differences between lymphoblastoid B cells and plasma B cells.

## Discussion

In this study, we investigated the 3D genome of aneuploid cancer and performed integrated analyses of Hi-C, WGS, and RNA-seq experiments in two MM cell lines. We found that aneuploid tumor Hi-C data are biased by CNVs and showed that the ICE algorithm can correct this CNV bias. Previous 3D genome studies of cancer also used ICE for normalization, but did not measure the influence of CNV bias[16, 17]. Recently a research group also designed a new method caICB, which based on ICE to eliminate CNV bias in cancer Hi-C data[43]. We found that the 3D cancer genome is influenced by cancer-specific genome alterations including CNVs and translocation events. Another recent study also proposes to use Hi-C data to detect cancer CNVs and translocations[44]. In this context, an interesting research direction is to integrate multiple types of genomic information, such as CNVs, translocations and 3D interactions to assemble more accurate cancer genomes, as has been done for normal cells[45, 46].

Recent studies have revealed that the interplay between higher-order chromatin structures and somatic copy-number alterations is important for cancer cell evolution[18–20]. Translocation frequency in cancer is highly correlated with the spatial proximity of two chromosome breakpoints in normal cells[18–20]. The 3D genome architecture such as pre-existing spatial proximity of two sites may contribute to translocation events[20]. Conversely, other studies also found that genomic variation events can rewire three-dimensional regulatory architectures and cause pathogenic phenotypes[47]. In addition, duplications within a TAD cause no changes in TAD structures, but duplications spanning a TAD boundary can result in formation of novel TADs and dysregulated gene expression during tissue development[48]. In this study, we find that 30% of the CNV breakpoints in MM occur at or near TAD boundaries, which may retain primary TAD structures and avoid ectopic enhancer-gene interactions[6]. The other 70% CNV breakpoints occur within the TAD structures, which may cause novel gene regulation in addition to the dosage effects of altered copy numbers[49]. Interestingly, TAD boundaries in normal B cells are also enriched by myeloma CNV breakpoints, suggesting that CNVs disrupting TAD

boundaries are selected against during cancer cell evolution. Both types of CNVs may be selected for their fitness impact during cancer cell evolution, and further studies are needed to understand their formation mechanisms and functional effects.

At the TAD scale, we found that MM genomes contain more TADs and the average TAD size is smaller than in normal B cells. This finding confirms previously reported results in prostate cancer[17]. RPMI-8226 and U266 cells have different numbers of chromosomes but similar numbers of TADs. Since the ICE method normalizes the Hi-C interactions to the per-copy level, CNVs may be responsible for the increased TAD numbers in MM cells compared to normal B cells. Based on previous studies and our findings, we propose that CNVs such as segmental duplications may generate more TADs, whose association with gene expression and functional consequences are worth further studying[48]. Heterogeneity of cancer cells may also contribute to more diverse 3D genomes within a cell population and therefore increase the detected TAD numbers. Single cell Hi-C techniques will help to confirm this possibility[50]. More studies of 3D cancer genomes will further clarify the relationship between genome alterations and 3D genome organization.

During cell differentiation, stimulation response, or cancer development, the 3D architecture of the genome is reorganized, which is associated with epigenetic alterations and changes in gene expression[16, 17, 51, 52]. In MM cells, we found that about 20% of genome regions switched between the A compartment and B compartment compared to normal B cells. This proportion of A/B compartment switching is higher than in breast cancer cells (12.4%, MCF10A vs. MCF7)[16]. Consistent with breast cancers, we found that switching of compartment type in MM is associated with changes in gene expression. Pathway enrichment analysis also showed that genes located in the switched compartments were strongly associated with the cytokine-cytokine receptor interaction pathway, hematopoietic cell lineage pathway and MAPK signaling pathway. We showed that genes in cytokine-cytokine receptor interaction pathway, including *IL1R1*, *IL1R2* and *MAP4K4* are downregulated and changed their 3D structures without copy number variations. *IL1R1* and *IL1R2* are two cytokine receptors which can bind with interleukin 1 and regulate the immune system and inflammation effects[53], and recently *IL-1* has been shown to enhance Th1-mediated immunity against cancer[54]. In the MAPK pathway, up to 50% MM patients have frequent mutations in genes of *NRAS*, *KRAS*, and *BRAF*[38]. Our results showed that genes *CACNG6*, *DUSP6* and *MAP4K3* in the MAPK pathway have reorganized spatial structures in MM, and associate with gene expression regulation as well as epigenetic mark changes, which might contribute to cancer development (Supplementary Fig. 7). Combining transcriptome and 3D genome analyses, we reveal that during MM development multiple levels of alterations such as CNVs, translocations, spatial genome reorganization occur and influence gene expression. However, since cancer types are diverse and alterations are heterogeneous, the phenomena observed in one cancer type may not hold in other cancer types. In the future, we need to further investigate whether these observations are universal phenomena across cancer types.

In summary, we have investigated the 3D genome of MM and analyzed the relationships among TADs, CNVs, translocations, and gene expression to identify MM-related pathways and key genes. These findings extend our understanding of MM as well as the spatial disorganization of aneuploid cancer genome, which may implicate in clinic treatment and drug development for MM.

## Methods

**Cell culture**. The RPMI8226 and U266 cell lines were purchased from ATCC (Virginia, USA). Cells were cultured in RPMI-1640 medium (ATCC-30-2001) with 10% fetal bovine serum and 5% $CO_2$ at 37 °C.

**Hi-C experiments**. Cells were grown to 70–80% confluence in 10-cm dishes, washed with phosphate-buffered saline and counted with a cell counting chamber. A total of $2–5 \times 10^6$ cells were isolated and cross-linked with 1% formaldehyde for 10 min at room temperature, and then 2.5 M-glycine solution was added to a final concentration of 0.2 M. Then cells were collected, flash-frozen in liquid nitrogen and stored at −80 °C. The Hi-C experiment was performed following the in situ Hi-C protocol[11]. Briefly, the cross-linked cells were lysed and digested with HindIII or MboI, filled with biotin-14-dATP, proximately ligated with T4 DNA ligase and reverse crosslinked with 5 M sodium chloride. Then the genome DNA was purified, sheared and size-selected. Biotin pull-down was performed to enrich target DNA fragments, followed by standard Illumina library construction.

**Whole-genome sequencing experiments and analysis**. Whole genome DNA of the two MM cell lines were extracted and sequenced at 50× depth through XTen (Illumina). The sequenced reads were mapped to the human reference genome (hg19) by the bwa-mem software[55]. Only uniquely mapped reads were used for downstream analysis. The Picard software (http://broadinstitute.github.io/picard) was used to remove PCR duplicates. The total mapping rate is above 90% for each sample.

**RNA-seq experiments and analysis**. Total mRNA with polyA tail was extracted and reverse transcribed to cDNA for sequencing. Three biological repeats were performed for each sample and 20 million reads was sequenced for each repeat. The sequenced reads were mapped to the human reference genome (hg19) by TopHat2[56] and gene expressions were quantified by Cufflinks[57]. We used the RStudio software for the downstream statistical analyses.

**Hi-C data analysis**. We performed reads mapping and filtering of the Hi-C data following previous methods[58]. Briefly, all Hi-C sequencing reads were mapped to the human reference genome (hg19) using Bowtie2[59]. The two ends of paired-end reads were mapped independently using the first 36 bases of each read. We filtered out redundant and non-uniquely mapped reads, and kept the reads within 500 bp upstream of enzyme cutting sites (HindIII or MboI) due to the size selection. We utilized the iterative correction and eigenvector decomposition (ICE) method[22] and HiCNorm[23] to normalize raw interaction matrices and compared their effect on correcting CNV bias.

**A/B compartment analysis**. We used ICE-normalized interaction matrices at 500-kb resolution to detect chromatin compartment types by R-package HiTC[36]. Positive or negative values of the first principal component separate chromatin regions into two spatially segregated compartments. The compartment with higher gene density was assigned as A compartments, and the other compartment was assigned as B compartment[16].

**TAD analysis**. We used ICE-normalized interaction matrices at 40-kb resolution to call TAD by a Perl script matrix2insulation.pl (http://github.com/blajoie/crane-nature-2015). A higher resolution was used because TADs are smaller than A/B compartments. Insulation Scores (IS) were calculated for each chromosome bin and valleys of IS identified TAD boundaries. TADs smaller than 200 kb or located in telomeres/centromeres were filtered out as in previous methods[37]. When comparing TADs between two cell lines, at least 70% overlap between two TADs were considered as conserved TADs[17]. We used Bedtools with the option of "intersectBed −f 0.70—r" to identify conserved TADs[60].

**CNV analysis**. CNVs were called by the Control-FREEC software at 40-kb and 500-kb resolution[61]. Uniquely-mapped reads of the WGS dataset or Hi-C data sets were used as inputs. The ploidy parameter for RPMI-8226 and U266 cells was set to 3 and 2, respectively, based on the karyotyping results. We also used the ploidy parameter of 2 for RPMI-8226 and found a similar CNV results as using the ploidy parameter 3.

**CNV bias correction in Hi-C data**. We defined a CNV block as a continuous region of chromosome with the same estimated copy number. For each Hi-C interaction matrix corresponding to a CNV block, the median diagonal interaction value was used to represent the interaction strength within that CNV block. We then fitted a linear regression model to assess the relationship between interaction strength and copy numbers of all CNV blocks. A significantly positive slope indicated the existence of CNV bias in Hi-C data of cancer samples. We applied this analysis on raw interaction matrices as well as ICE and HiCNorm corrected interaction matrices to evaluate the correction of CNV bias in Hi-C data.

**Translocation analysis**. The WGS reads were used to identify translocation events by the CREST software with the default parameters[24]. Further filtering criteria included (i) the supporting reads at both sites of an inter-chromosomal translocation are greater than 10% of the total reads at these positions; (ii) the sum of supporting reads at the two sites of an inter-chromosomal translocation are greater than 40. We used Fisher's exact test to check whether the inter-chromosomal

translocations identified from WGS data were enriched in the top 100 bin pairs with the highest inter-chromosomal Hi-C interactions.

**Overlap between TAD boundaries and CNV breakpoints**. The cancer CNV dataset was downloaded from http://202.97.205.78/CNVD/, a database of "Copy Number Variation in Disease" via manual text mining from published papers. This database includes 845 diseases across 20 species with half records for human. Only CNVs of MM were selected which includes 302 instances. CNVs with length larger than 500 kb and sample rate over 0.1 were kept. After filtering, there remains 161 CNVs. We then plotted the insulation scores of GM12878 within 1 Mb regions from these filtered CNV breakpoints or GM12878 TAD boundaries.

**Gene ontology analysis**. Gene pathway enrichment analysis for the switched on/off genes used the DAVID Bioinformatics Resources 6.7[62]. All the human genes were used as the background gene list. Switched on genes are defined as their genomic compartment changed from B to A in both MM cell lines (compared with GM12878). Switched off genes are defined as their genomic compartment changed from A to B in both MM cell lines (compared with GM12878).

**ChIP-seq data analysis**. We obtained the ChIP-seq data of U266 cells from the NCBI BioProject database (PRJEB1912/ERS333898), and the ChIP-seq data of GM12878 cells from the UCSC Table Browser. The raw fastq files of U266 ChIP-seq data were processed using cutadapt[63] software to remove adaptor sequence, then reads were mapped by bowtie2[59] using reference hg19, and duplicated reads were removed by picard tools (http://broadinstitute.github.io/picard). The mapped bam files were transferred to the MACS[64, 65] software to call peaks with default parameters as the ENCODE pipeline recommended.

**Statistical tests**. We calculated the p-values of Fig. 4b, f to check whether the overlap is significant by comparing the overlap between different cell types using Fisher's exact test.

**Data availability**. All essential codes used for analysis are available at GitHub (http://github.com/ChengLiLab/myeloma). The Gene Expression Omnibus (GEO) accession number for the WGS, RNA-seq, and Hi-C data sets generated in study is GSE87585. The WGS dataset, RNA-seq dataset, and Hi-C dataset of GM12878 can be accessed by ERX069505 (SRA), GSM758560 (GEO), and GSE63525 (GEO) via NCBI. The CNVD data can be accessed via http://202.97.205.78/CNVD/.

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

## Acknowledgements

We thank Chengran Xu, Ge Gao, Lichen Ren, Yang Chen, and Juntao Gao for their help on Hi-C experiments, and Yujie Sun, Yue Huang, and reviewers for critical suggestions. This work was supported by funding from the Peking-Tsinghua Center for Life Sciences, the School of Life Sciences and Center for Statistical Science of Peking University, the National Natural Science Foundation (China Key Research Grant 71532001), and the Chinese National Key Projects of Research and Development (2016YFA0100103).

## Author contributions

C.L. designed and supervised this project. T.L., L.J., Q.C., and Y.Y. carried out the Hi-C, WGS, and RNA-seq experiments. R.L. and P.Z. built the pipeline for Hi-C data analysis. P.W. and R.L. performed integrative data analysis. Y.L. and D.T. participated in data analysis. P.W., R.L., T.L., and C.L. wrote the manuscript. All authors approved the manuscript.

## Additional information

**Competing interests:** The authors declare no competing financial interests.

