## [Peer Review File · Nature Communications]

Reviewers' comments:

Reviewer #1 (Remarks to the Author):

Wu et al convincingly demonstrate that copy number variations are associated with TAD boundaries in multiple myeloma. If there are 4 copies of a particular genomic region and the adjacent region has 2, the point of transition will result in a TAD boundary. This is intuitive as one would expect higher degrees of local interaction within regions that have more copies. Upon normalization each region will appear to have similar levels of interaction. It appears based on the extent of copy number variations overlapping with TAD boundaries that not all CNV are associated with altered TAD boundaries (30.7%). It would be interesting to determine why this is not the case at some CNV but not others. This point should be raised in the discussion. The authors also demonstrate alterations in A and B compartments corresponding to altered gene expression (with relevant and interesting pathways found within altered compartments). While this work is well done, it is very similar to previous work done in prostate cancer and therefore the novelty is reduced. Despite this, it is important to further investigate as our knowledge in this field is extremely limited.

An inherent limitation of Hi-C, and indeed any high throughput sequencing technique, is that it is an average of millions of cells. Therefore any interactions found may be within subpopulations. For example in figure 3 (I don't have the paper in front of me right now), they show a Hi-C matrix wherein the TAD boundaries correspond to copy number variations. There appears to be a high frequency interaction between TADs on either side of a CNV. Is this representative of those chromosomes which have higher copy numbers or those with less? It is suggested that the authors perform FISH for such regions of interest.

Reviewer #2 (Remarks to the Author):

Wu et al. aim to investigate the 3D genome of two multiple myeloma cell lines by performing an integrated analyses of Hi-C, WGS, and RNA-seq experiments. They stated that the breakpoints of copy-number abnormalities overlap with topologically associating domain (TAD) boundaries, suggesting that TAD boundaries facilitate the formation of copy-number abnormalities breakpoints. Furthermore, they suggest that the number of TADs and changes in expression are found in tumor cells compared with normal B-cells and those changes are associated with the 3D distribution of the tumor genome. The study uses cutting-edge technologies and, as far as I know, it is the first of the kind in MM. Although these types of approaches are promising in the study of tumor genome architecture, my overall feeling is that the data have not been studied in detail and the authors just scratched the surface of the data.

They performed comparisons between 2 MM and a lymphoblastoid cell line (control). They performed comparisons between MM and control, however the comparison between MM cell lines is limited. That should be further explored, especially considering the different genetic background of them.

Furthermore, a more detail section explaining the statistical testes performed in the comparisons is desired.

When comparing the genome regions that switched between the A compartment and B compartment, the authors considered a lymphoblastoid cell line as a normal B cells. I would be more careful about that statement. Lymphoblastoid cell lines are EBV transformed and highly proliferative so many expression changes are expected compared with a normal B cell.

To be more conclusive, the analysis of a larger number of cell lines and primary MM cases (if feasible) would be desired.

The analysis of genes/pathways up- and underregulated needs to be further explored. The authors integrated Hi-C, WGS, and RNA-seq data, however they did not consider mutation data, which

may affect some of their results. For example, the authors highlighted the MAPK signaling is one of the pathways associated with changes in gene expression. Interestingly, both U266 and RPMI8226 have activating mutations in genes of the pathway (BRAF p.K601N and KRAS p.G12A, respectively). The results should be put in the context considering the mutation profile of each cell line.

Minor comments:

Why do they show 2 karyotypes per cell line in figure S1?

Figure S2e legend is missing

An additional proofreading is suggested, as there are several typos in the text

Reviewer #3 (Remarks to the Author):

The authors present a novel and interesting way of analysing Hi-C data together with WGS data. Especially, showing that Hi-C data is strongly biased by copy number changes and identifying the best method to correct for such bias is very important. The data presented show that copy number changes are associated with a disruption of the 3D genome architecture and that gene expression changes in cancer may be related to 3D genome conformation. This is not novel and has been shown previously. The novelty is more that the authors show that Hi-C data can be used to confirm translocations from WGS data and to identify novel MM-related key genes and regions. However there are a number of outstanding concerns that need to be addressed to clarify and substantiate the results.

Specific Concerns:

1. Inter-chromosomal interactions and translocations.

- The overlap between RPMI-8226 translocations and top Hi-C trans interactions is very small (4 translocations), even though the Fisher's exact test p-value is significant. This association needs to be confirmed at the genome-wide level for example by comparing the observed overlap between trans interactions and WGS translocation to a random overlap (O/E fold change)?
- How many of the newly identified inter-chromosomal interactions were common between the two MM cell lines? Are they present in the normal GM12878 Hi-C data?
- It is important to show that the newly identified in WGS and Hi-C data translocations are resulting in aberrant expression of cancer-related genes located at these regions. Are genes affected by the newly identified translocations in MM cells differentially expressed compared to cells that do not harbour these translocations e.g. normal B-cells?

2. CNVs and TAD boundaries.

- Information regarding number of identified TADs in each cell line and their sizes should be included in this section (currently lines 175 – 185).
- To study the association between TAD boundaries and CNV breakpoints, the authors' averaged the insulation scores within 1Mb of all TAD boundaries. Considering the median size of TADs in these cells is under 1Mb, using this approach could result in adjacent TAD boundary being included as a surrounding region and therefore affecting the insulation profile. This needs to be addressed.
- The authors need to address if the boundaries of TADs that are specific to MM cells (740 TADs) are enriched for CNVs as compared to boundaries of conserved TADs or TADs in GM12878 cells?

3. In order to verify whether TAD boundaries are more susceptible to CNVs compared to the rest of the TAD, the authors used a database on cancer CNVs and TAD boundaries in normal B cells (GM12878).

- Information regarding how the GM12878 data was obtained and processed should be included in the paper with a reference if public data was used.
- It is not clear how many cancer CNVs were included in the analysis?

- What is the percentage overlap between cancer CNVs and TAD boundaries in GM12787 cells?
4. "We defined two TADs as being conserved between two samples if they overlapped by more than 70% of the TAD regions" (lines 179 – 180). Using this approach some changes in TAD architecture could be potentially missed, e.g. if large TAD in one sample is sub-divided into multiple smaller TADs in other sample, that would result in some of the sub-TADs being considered as conserved as > 70% of a TAD overlap? Looking at the overlap between TAD boundaries could be more appropriate here.
5. Lines 175 – 184 describe changes in TAD organization, but do not include any results in relation to gene expression changes and therefore do not fit into section "3D genome reorganization in multiple myeloma is associated with gene expression change".
6. A and B compartments.
- The authors need to address what is the relationship between TADs and A/B compartments? Do compartment boundaries overlap with TAD boundaries?
 - In regions where there is a switch between A and B compartment compared to normal cells, is there also a change in TADs?
7. Pathway enrichment analysis for genes located at regions that switch between compartment A and B have identified some interesting MM-related and cancer-related pathways.
- Are these genes differentially expressed between MM cell lines and GM12878 cells? Are these changes consistent with the direction of change between A and B compartments?
 - Fig.5b and c shows changes in ChIP-seq marks between normal and MM cells, however these are not discussed in enough detail in text or in the figure legend.
 - How the ChIP-seq data has been obtained? If public data has been used, a reference should be included.
8. Fig. 5b and c.
- It is not clear what region of the genome is shown on this example. Genomic location and scale should be added.
 - The insulation score profile looks flat and the dips at TAD boundaries are not really visible – would changing the scale of the score help?
 - TADs and A/B compartments should be clearly marked on the figure for an easier interpretation.
9. Authors haven't presented any results relating to local intra-chromosomal interactions in each of the cell types. Are local interactions affected by CNVs? Does the interaction profile of a TAD changes when a CNV is present at the boundary? Is there an association with gene expression changes?
10. Discussion
- Statement "suggesting that TAD boundaries may promote genome rearrangements" (line 223 – 224) is not supported by the presented results. Also, in next paragraph authors propose that "CNVs and chimeric chromosomes may cause more TADs by reorganizing the 3D architecture of the genome", which suggests an opposite effect?
 - "RPMI-8226 and U266 cells have different numbers of chromosomes, but similar number of TADs". Isn't that expected, considering that TADs are called using ICE-normalized data that accounts for copy number changes?
 - The summary paragraph is too vague and unfocused.
11. In the Materials and Methods part of the paper, authors should include information about other datasets used in this study, e.g. GM 12787 Hi-C and RNA-seq and ChIP-seq data.

Response to Reviewers

We thank the reviewers and editor for the critical comments, which have greatly helped us improve the results and manuscript. Please see below for our point-to point responses. Text changes in the revised manuscript are indicated in red color.

Reviewer #1 (Remarks to the Author):

Wu et al convincingly demonstrate that copy number variations are associated with TAD boundaries in multiple myeloma. If there are 4 copies of a particular genomic region and the adjacent region has 2, the point of transition will result in a TAD boundary. This is intuitive as one would expect higher degrees of local interaction within regions that have more copies. Upon normalization each region will appear to have similar levels of interaction. It appears based on the extent of copy number variations overlapping with TAD boundaries that not all CNV are associated with altered TAD boundaries (30.7%). It would be interesting to determine why this is not the case at some CNV but not others. This point should be raised in the discussion.

Answer: In this study, we have found significant overlapping between CNV breakpoints and TAD boundaries. We also notice that not all CNV breakpoints coincide with TAD boundaries and some are inside TADs. This is an interesting question that we have added the following discussion in the discussion part of the manuscript (page 10, citations in the main text):

“Recent studies have revealed that the interplay between higher-order chromatin structures and somatic copy-number alterations are important for cancer cell evolution. Translocation frequency in cancer is highly correlated with the spatial proximity of two chromosome breakpoints in normal cells. The three-dimensional genome architecture such as pre-existing spatial proximity of two sites may contribute to translocation events. However, other studies also found that genomic variation events can rewire three-dimensional regulatory architectures and cause pathogenic phenotypes. In addition, duplications within a TAD cause no changes in TAD structures, but duplications spanning a TAD boundary can result in formation of novel TADs and dysregulated gene expression during tissue development. In this paper, we find that 30% of the CNV breakpoints in MM occur at TAD boundaries, which may retain primary TAD structures and avoid ectopic

enhancer-gene interactions. The other 70% CNV breakpoints occur within the TAD structures, which may cause novel gene regulation in addition to the dosage effects of altered copy numbers. Both types of CNVs may be selected for their fitness impact during cancer cell evolution, and further studies are needed to understand their formation mechanisms and functional effects.”

The authors also demonstrate alterations in A and B compartments corresponding to altered gene expression (with relevant and interesting pathways found within altered compartments). While this work is well done, it is very similar to previous work done in prostate cancer and therefore the novelty is reduced. Despite this, it is important to further investigate as our knowledge in this field is extremely limited.

Answer: In this study, we find that compartment A/B switches are associated with gene expression changes in two multiple myeloma cells, consistent with previous findings in prostate cancer and breast cancer. However, whether this conclusion holds true in other cancer types is still not known. Also, how many regions have compartment A/B switches and how many associated genes are differentially expressed in other cancer types is still not known. Our findings suggest that the association between compartment A/B switches and gene expression changes may be a general rule in many cancer types. However further studies about other cancers are needed to solidify this conclusion.

We have added the following discussion in page 12 of the manuscript:

“Our study extends previous findings to multiple myeloma. However, since cancer types are diverse and alterations are heterogeneous, the phenomena observed in one cancer type may not hold in other cancer types. The chromosome regions that have switched compartment A/B and the differentially expressed genes associated with the switching are different between MM cells and other cancer types. In the future, we need to further investigate whether these observations are universal phenomena across cancer types.”

An inherent limitation of Hi-C, and indeed any high throughput sequencing technique, is that it is an average of millions of cells. Therefore, any interactions found may be within subpopulations.

Answer: We agree. In Figures S1a, S1b, S1c, and S1d, both RPMI-8226 and U266 cell lines have various karyotypes at the single cell level, but the dominant karyotype of RPMI-8226 is near triploid and that of U266 is near diploid. In this study, we correlate the average level of chromatin interactions in many cells with the average level of CNVs and gene expression. With the development of single-cell Hi-C techniques, we expect to explore the heterogeneity of the 3D genome structures and its association with single-cell genome and transcriptome data.

For example, in figure 3 (I don't have the paper in front of me right now), they show a Hi-C matrix wherein the TAD boundaries correspond to copy number variations. There appears to be a high frequency interaction between TADs on either side of a CNV. Is this representative of those chromosomes which have higher copy numbers or those with less? It is suggested that the authors perform FISH for such regions of interest.

Answer: We found a significant overlapping between CNV breakpoints and TAD boundaries. We used Figure 3a to show a typical example (the original Figure 3a title "RPMI-8226, chr5: 80Mb-92Mb" has been corrected to "RPMI-8226, chr15: 80Mb-92Mb"). In this example, there are higher interactions between the chromatin regions surrounding one deletion CNV (arrow in Figure 3a), raising the general possibility that deletion CNV promotes higher interactions between the surrounding chromatin regions.

We explored this possibility from two aspects. First, the high interactions between TAD (chr15:82.56Mb-83.20Mb) and TAD (chr15:84.88Mb-85.84Mb) not only exist in the two MM cells (Figure 3a), but also exist in the GM12878 normal B cells which have no CNVs in this region (Figure R1), suggesting that this high interaction is less likely caused by CNVs. Second, for each CNV gain or CNV loss in RPMI-8226, we first identified the nearest TADs on either side of the CNV, then used the median interaction between the two TADs to measure their interaction strength. The distribution of the TAD-TAD interactions surrounding CNVs is not significantly different from those surrounding randomly selected non-CNV chromosome regions (Figure R2). We therefore conclude that CNVs do not significantly alter interactions between surrounding chromosome regions. Our lab currently does not have FISH experimental setup, but we plan to perform FISH experiments with the help from collaborators and use more cancer Hi-C samples to revisit this issue in future studies.

Fig. R1 Interaction heatmap of chromosome15: 80 Mb – 92 Mb in GM12878 cells.

Fig. R2 Boxplot of TAD-TAD interactions surrounding CNVs.

Reviewer #2 (Remarks to the Author):

Wu et al. aim to investigate the 3D genome of two multiple myeloma cell lines by performing an integrated analyses of Hi-C, WGS, and RNA-seq experiments. They stated that the breakpoints of copy-number abnormalities overlap with topologically associating domain (TAD) boundaries, suggesting that TAD boundaries facilitate the formation of copy-number abnormalities breakpoints. Furthermore, they suggest that the number of TADs and changes in expression are found in tumor cells compared with normal B-cells and those changes are associated with the 3D distribution of the tumor genome. The study uses cutting-edge technologies and, as far as I know, it is the first of the kind in MM. Although these types of approaches are promising in the study of tumor genome architecture, my overall feeling is that the data have not been studied in detail and the authors just scratched the surface of the data.

Answer: We thank the reviewer for the positive comments. Currently there are only a few published studies about cancer 3D genomes. Our work has generated the first 3D genome architectures of multiple myeloma using cell lines. Although the data have helped us understand the MM cancer from the viewpoint of 3D genome structures and its correlation

with gene expression and CNVs, we agree with the reviewer that more analyses can be done, such as comparative studies of 3D genomes of multiple cancer types by combining public data and our data.

They performed comparisons between 2 MM and a lymphoblastoid cell line (control). They performed comparisons between MM and control, however the comparison between MM cell lines is limited. That should be further explored, especially considering the different genetic background of them.

Answer: We agree that the different genetic background of the cell lines can complicate the comparison of their 3D genome structures. To validate the similarities of the cell lines that we use, we clustered the 3 cell lines with other 4 cell lines (PC3 and PrEC: prostate cancer; MCF7 and MCF10A: breast cancer) by using the first principal component (PC1) of their Hi-C interaction matrix data, and the results showed that same type of cancer cell lines clustered together well (Fig. R3). We modified Fig. 4A to include the A/B compartment information of the extra 4 cell lines.

Cluster Dendrogram

Fig. R3 Clustering of 7 cell lines according to their A/B compartment information. Clustering was performed by using the PC1 of the bins which ranked the first 50% in all the genome with hclust method in R.

Furthermore, a more detail section explaining the statistical testes performed in the comparisons is desired.

Answer: We have added a paragraph to explain the statistical tests we used for the comparison of TAD conservation at the Method section in page 15 and 23.

When comparing the genome regions that switched between the A compartment and B compartment, the authors considered a lymphoblastoid cell line as a normal B cells. I would

be more careful about that statement. Lymphoblastoid cell lines are EBV transformed and highly proliferative so many expression changes are expected compared with a normal B cell.

To be more conclusive, the analysis of a larger number of cell lines and primary MM cases (if feasible) would be desired.

Answer: Since the RPMI-8226 and U266 cell lines are widely used as model cell lines in multiple myeloma research, we used these two cell lines in our study. Normal cell lines for blood lineage cancers are difficult to obtain because once developed, these cancer types spare few normal cell counterparts. Both of these MM cell lines as well as many other cancer cell lines lack appropriate control cells, which should be the normal tissue cells from the original patients. In this study we included the GM12878 cells (a B lymphoblastoid cell) for the control purpose, which may have some deviations from the ideal controls (the plasma B cell). Due to the limitation of clinical MM samples and experimental costs, we only studied two MM cell lines here. Nevertheless, the findings from this study are good rationale for designing and justifying larger 3D genome studies of more MM clinical samples and cell lines in the future. We additionally analyzed the A/B compartment of 4 cell lines of other cancer types using their public data (Fig. 4A). MCF7 and MCF10A are breast cancer cell lines, and PC3 and PrEC are prostate cancer cell lines.

The analysis of genes/pathways up- and under-regulated needs to be further explored. The authors integrated Hi-C, WGS, and RNA-seq data, however they did not consider mutation data, which may affect some of their results. For example, the authors highlighted the MAPK signaling is one of the pathways associated with changes in gene expression. Interestingly, both U266 and RPMI8226 have activating mutations in genes of the pathway (BRAF p.K601N and KRAS p.G12A, respectively). The results should be put in the context considering the mutation profile of each cell line.

Answer: Thanks for the suggestion. To explore this question, we first detected the SNVs of RPMI 8226 and U266 cell lines with samtools/bcftools, and then annotated these genes that locate in A/B switched regions in both cell lines with their mutation information. 911 of the 1157 genes that switched from B to A compartment have no protein-coding mutations, and 164 of the 208 genes that switched from A to B compartment have no protein-coding

mutations. We used these genes without protein-coding mutations (1119) for further pathway analysis and the Fig. 5A and its legend were updated.

Minor comments:

Why do they show 2 karyotypes per cell line in figure S1?

Answer: Cancer cells usually have heterogeneous sub-clones that have non-identical karyotypes. From the karyotype experiments, we found that both the RPMI-8226 and U266 cell lines contain a mixture of sub-clones, and showed two representative karyotypes of each cell line as examples. This also means that WGS, RNA-seq and Hi-C data are averaged results of heterogeneous cells. In the future, single-cell analysis of transcriptome and genomes are needed to better reveal the heterogeneity of the cancer cells and its functional consequences.

Figure S2e legend is missing

Answer: Thanks for pointing this out. This figure quantifies the correlation coefficients of contact matrix in Figure S2d. Since the contact matrix of Hind III treatment and Mbol treatment in RPMI-8226 are similar to each other, we combined the two datasets to achieve higher number of valid sequence reads.

We have added the Figure S2e legend as follows (page 24): (e) Left: Comparing interaction scores of chromosome 1 with RPMI-8226 (Hind III) at y-axis and RPMI-8226 (Mbol) at x-axis; 200 kb resolution; the Pearson's correlation is 0.89; Right: Comparing interaction scores of whole genome with RPMI-8226 (Hind III) at y-axis and RPMI-8226 (Mbol) at x-axis; 3 Mb resolution; the Pearson's correlation is 0.87.

An additional proofreading is suggested, as there are several typos in the text

Answer: Thanks for pointing this out. We have corrected typos in spelling and grammar and revised sentences to state our conclusions more clearly, and marked them in red in the revised manuscript.

Reviewer #3 (Remarks to the Author):

The authors present a novel and interesting way of analyzing Hi-C data together with WGS data. Especially, showing that Hi-C data is strongly biased by copy number changes and identifying the best method to correct for such bias is very important. The data presented show that copy number changes are associated with a disruption of the 3D genome architecture and that gene expression changes in cancer may be related to 3D genome conformation. This is not novel and has been shown previously. The novelty is more that the authors show that Hi-C data can be used to confirm translocations from WGS data and to identify novel MM-related key genes and regions. However there are a number of outstanding concerns that need to be addressed to clarify and substantiate the results.

Answer: we thank the reviewer for the constructive comments and please see our responses below.

Specific Concerns:

1. Inter-chromosomal interactions and translocations.

- The overlap between RPMI-8226 translocations and top Hi-C trans interactions is very small (4 translocations), even though the Fisher's exact test p-value is significant. This association needs to be confirmed at the genome-wide level for example by comparing the observed overlap between trans interactions and WGS translocation to a random overlap (O/E fold change)?

Answer: Since the Hi-C trans-interaction matrices are sparse with over 80% zeros, we used the 5 Mb windows to find the top 100 highest trans-interactions. These top 100 trans-interactions not only contain translocation events, but also inter-chromosomal chromatin

Fig. R4. Overlap between random sites and translocation sites in RPMI 8226 cells.

interactions. To assess the degree of random overlapping, we applied a random sampling analysis in the RPMI-8226 cell line by generating 100 random trans-interaction sites and calculating the overlapped sites with the WGS translocations. Over 1000 random samplings, the expected overlaps following a skewed distribution with a mean of 0.044 as shown in Figure R4. The O/E fold change is 90.91, indicating that the overlap between translocations and top Hi-C trans-interactions is significant.

We added Fig. R4 in the manuscript as Fig. S4a, and updated the text in page 6 and related figure legends.

• How many of the newly identified inter-chromosomal interactions were common between the two MM cell lines? Are they present in the normal GM12878 Hi-C data?

Answer: We calculated inter-chromosomal interactions across the whole genome with a 1Mb bin-size and the result showed that only the top few inter-interactions showed significant stronger interaction signal (Fig. R5). We select the top 100 inter-chromosomal interactions as real inter-interactions, and 73 of them were common between the two MM cell lines. Among the 73 common inter-interactions in MM cells, 67 of them are also present in the normal GM12878 Hi-C data, suggesting conserved inter-chromosomal interactions among B cells.

We added Fig. R5 in the manuscript as Fig. S5f, and updated the text in page 6 and related figure legends.

Fig. R5 Density plot for top 1000 inter-chromosomal interactions in three cell lines. Dashed lines indicate 10% cutoff

- It is important to show that the newly identified in WGS and Hi-C data translocations are resulting in aberrant expression of cancer-related genes located at these regions. Are genes affected by the newly identified translocations in MM cells differentially expressed compared to cells that do not harbor these translocations e.g. normal B-cells?

Answer: We agree that it is important to investigate the functional consequences of translocation events. In RPMI-8226, among the 7 sites of translocations that have high Hi-C interactions, there are 105 genes within 1 Mb distance from the translocation sites. Among the 105 genes there are 52 differentially expressed genes between RPMI-8226 and GM12878, including cancer-related genes such as MYC, WWOX, MADD etc. This gene list and expression changes are listed as Table S2. In U266, there are 63 genes within 1 Mb regions near the 3 sites of translocations, and 33 of these genes are differentially expressed compared to GM12878. We updated these results in page 6 of the manuscript.

2. CNVs and TAD boundaries.

- Information regarding number of identified TADs in each cell line and their sizes should be included in this section (currently lines 175 – 185).

Answer: We have added the following description of the identified TADs in the manuscript text (page 8):

We next compared TADs in the MM cells and B cells. We called TADs from Hi-C interaction matrices at the 40-kb resolution and identified 2756, 3457 and 3342 TADs in GM12878, RPMI-8226 and U266 cells, respectively (Figure 4f), and their median TAD sizes are 800 kb, 600 kb and 640 kb, respectively (Figure 4e).

- To study the association between TAD boundaries are CNV breakpoints, the authors' averaged the insulation scores within 1Mb of all TAD boundaries. Considering the median size of TADs in these cells in under 1Mb, using this approach could result in adjacent TAD boundary being included as a surrounding region and therefore affecting the insulation profile. This needs to be addressed.

Answer: Thanks for pointing this out. In accordance with the reviewer's suggestion that the averaged insulation scores within 1Mb could stride over neighboring TAD boundaries, we notice that the averaged insulation scores first increased away from the CNV breakpoints, and then decreased slightly (Figure 3c). We did not change our results due to two reasons: first, the effect on the insulation profile is weak as the results are significant compared to random sites; second, similar analyses have been applied in studying the enrichment of ChIP-seq signals at specific genomic positions such as CTCF in TAD boundaries¹.

- The authors need to address if the boundaries of TADs that are specific to MM cells (740 TADs) enriched for CNVs as compared to boundaries of conserved TADs or TADs in GM12878 cells?

Answer: Among the boundaries of 740 specific TADs in MM cells (Figure 4f), 124 of them were CNV breakpoints of RPMI 8226 cells; 250 boundaries of 1281 conserved TADs were CNV breakpoints in RPMI 8226 cells; and 133 of boundaries of 672 specific TADs in GM12878 cells were CNV breakpoints of RPMI 8226 cells ($p = 0.24$, proportions test).

3. In order to verify whether TAD boundaries are more susceptible to CNVs compared to the rest of the TAD, the authors used a database on cancer CNVs and TAD boundaries in normal B cells (GM12878).

- Information regarding how the GM12878 data was obtained and processed should be included in the paper with a reference if public data was used.

Answer: The GM12878 Hi-C data were downloaded from GEO (accession number GSE63525), and the analysis was performed similarly as MM Hi-C data, as described in Hi-C data analysis section. We added this information in the "Code and data" section of materials and methods (page 15).

- It is not clear how many cancer CNVs were included in the analysis?

Answer: We used 302 CNVs occurred in MM from a manually curated cancer CNV database (CNVD). We have included a method description about this analysis in page 15 as following:

“The cancer CNV dataset was download from CNVD (<http://bioinfo.hrbmu.edu.cn/CNVD>), a database of Copy Number Variation in Disease via manual text mining from published papers. 302 CNVs occurred in multiple myeloma were first selected from CNVD, and 161 CNVs with length larger than 500 kb and sample frequency over 10% were kept. We then plotted the average insulation scores within 1 Mb of these filtered CNV breakpoints based on the GM12878 Hi-C data.”

• What is the percentage overlap between cancer CNVs and TAD boundaries in GM12878 cells?

Answer: Among the 322 CNV breakpoints of the filtered 161 CNVs, there are 98 breakpoints within the distance of 80 kb to GM12878 TAD boundaries. In contrast, for a randomly sampled 322 40-kb bins across whole genome, only 22 bins are within 80 kb of the GM12878 TAD boundaries ($p = 3.196e-14$, proportions test).

4. “We defined two TADs as being conserved between two samples if they overlapped by more than 70% of the TAD regions” (lines 179 – 180). Using this approach some changes in TAD architecture could be potentially missed, e.g. if large TAD in one sample is sub-divided into multiple smaller TADs in other sample, that would result in some of the sub-TADs being considered as conserved as > 70% of a TAD overlap? Looking at the overlap between TAD boundaries could be more appropriate here.

Answer: In our analysis, we used the insulation score profiles to define TAD boundaries at the 40 kb resolution. The length distribution of TADs showed two peaks, one is at 120 kb and the other is at 600 kb (Fig. R6). We then filtered out those TADs with length smaller than 200 kb. For the TADs with length larger than 200 kb, we compared them with the nearest TADs in another sample. If the two TADs overlap with more than 70% of either TAD, we define them as conserved TADs. Therefore, the situations that some of the sub-TADs being considered as conserved are avoided.

We have corrected the text into “We defined two TADs as being conserved between two samples if they overlapped by more than 70% of the TAD regions reciprocally.”

Fig. R6 Distribution of length of TADs in U266 cells.

5. Lines 175 – 184 describe changes in TAD organization, but do not include any results in relation to gene expression changes and therefore do not fit into section “3D genome reorganization in multiple myeloma is associated with gene expression change”.

Answer: We changed the section title to “3D genome reorganization in multiple myeloma and association between A/B compartment switches and gene expression changes” (page 8). Similar to previous findings in breast cancer, we found that A/B compartment switches between normal and cancer cells are associated with gene expression changes in the expected direction (Fig.4C).

6. A and B compartments.

- The authors need to address what is the relationship between TADs and A/B compartments? Do compartment boundaries overlap with TAD boundaries?
- In regions where there is a switch between A and B compartment compared to normal cells, is there also a change in TADs?

Answer: We performed analysis following these suggestions. 54.8%, 64% and 61% of the compartment boundaries overlap with TAD boundaries in GM12878, RPMI 8226 and U266 cells separately, which are all significant comparing to simulated data (Fig.R7), demonstrating that compartment boundaries prefer to overlap with TAD boundaries. In addition, 42% and 40.5% of the regions that switch between A or B compartment compared to normal cells in U266 and RPMI 8226 cells, respectively, also have a change in TADs, which are not significant comparing to simulation data (Fig. R8).

Fig. R7 Percentage of compartment boundaries that overlapped with TAD boundaries in RPMI 8226, U266 and GM12878 cell lines. Blank lines were density plots of simulated data for 1000 times.

7. Pathway enrichment analysis for genes located at regions that switch between compartment A and B have identified some interesting MM-related and cancer-related pathways.

- Are these genes differentially expressed between MM cell lines and GM12878 cells? Are these changes consistent with the direction of change between A and B compartments?

Answer: In the pathway enrichment analysis (Figure 5A), we used only the genes that are both differentially expressed between MM cell lines and GM12878 and have concordant change direction as A/B switching (i.e. up-regulation for B to A switching).

- Fig.5b and c shows changes in ChIP-seq marks between normal and MM cells, however these are not discussed in enough detail in text or in the figure legend.

Answer: We have included details in the figure legend as following (page 24):

H3K4me3 is annotated as promoters, H3K4me1/H3K27ac as active enhancers, and H3K36me3 as transcribed gene body and H3K9me3 as heterochromatin.

- How the ChIP-seq data has been obtained? If public data has been used, a reference should be included.

Answer: Thanks for pointing this out. We have added the following descriptions in page 15:

ChIP-seq data analysis: We obtained the ChIP-seq data of U266 cells from the NCBI BioProject database (PRJEB1912/ERS333898), and the ChIP-seq data of GM12878 cells from the UCSC Table Browser. The raw fastq files of U266 ChIP-seq data were processed using cutadapt61 software to remove adaptor sequence, then reads were mapped by bowtie2 at reference hg19, duplicated reads were removed by picard tools (<http://broadinstitute.github.io/picard>). The mapped bam files were transferred to the MACS software to call peaks with default parameters as the ENCODE pipeline recommended.

8. Fig. 5b and c.

- It is not clear what region of the genome is shown on this example. Genomic location and scale should be added.

Answer: Thanks for pointing this out. In Figures 5b and 5c, we showed several genes from the IL receptor family which have differential gene expressions and nearby TAD boundary changes, but no copy number variations. We have added the genomic location and scales to the Fig 5b and 5c, and the figure legend was updated.

- The insulation score profile looks flat and the dips at TAD boundaries are not really visible – would changing the scale of the score help?

Answer: Thanks for the suggestion. We enlarged the Y-axis scale of the insulation score to make it more visible.

- TADs and A/B compartments should be clearly marked on the figure for an easier interpretation.

Answer: Thanks for pointing this out. We marked the A/B compartments and added dashed lines indicating the boundaries of TADs.

9. Authors haven't presented any results relating to local intra-chromosomal interactions in each of the cell types. Are local interactions affected by CNVs? Does the interaction profile of a TAD change when a CNV is present at the boundary? Is there an association with gene expression changes?

Answer: Following this suggestion, we checked that if local interactions are affected by CNVs from TADs level. In U266 cells, the original TADs number were 4089, we filtered out the TADs with length below 160 kb, which were not of typical TADs length and may stand for non-TAD regions. After filtering, there are about ~3300 TADs. We used the median interaction score within a TAD to estimate its local interactions, and the median copy number of the bins in that TAD to stand for its copy number. We found that local interactions are correlated with copy numbers in ICE normalized interaction matrix at the TAD level. These results suggest that in the normalized interaction matrices, the CNV did affect the local interactions within a TAD.

We next asked whether local interactions between two adjacent TADs are changed when a CNV is present at the boundary. We used the median interactions between two adjacent TADs as their local interactions and found this value is increased when there is a copy number gain across the boundary, in the ICE normalized matrix. The same as gene expression(Fig.R10).

Fig. R9 Local interactions within a TAD (U266 ICE Hi-C matrix 40 kb).

Fig. R10 Gene expression within a TAD (U266).

10. Discussion

- Statement “suggesting that TAD boundaries may promote genome rearrangements” (line 223 – 224) is not supported by the presented results. Also, in next paragraph authors propose that “CNVs and chimeric chromosomes may cause more TADs by reorganizing the 3D architecture of the genome”, which suggests an opposite effect?

Answer: Based on published studies and our results, we propose that the 3D genome and CNVs influences each other. On one hand, CNVs that disrupt TAD boundaries in normal cells are selected against if they cause too much gene expression dysregulation, leading to the observed enrichment of CNV breakpoints at TAD boundaries in normal cells. On the other hand, some CNVs occurred in a cancer cells can promote cell growth and propagate through cancer cell proliferation, and these CNVs may create more TADs (e.g. segmental duplication covering and TAD boundary creates a neo-TAD). We have revised the above two sentences as below to better reflect these ideas:

“Interestingly, TAD boundaries in normal B cells are also enriched by myeloma CNV breakpoints, suggesting that CNVs disrupting TAD boundaries are selected against during cancer cell evolution.” (page 10)

“Based on previous studies and our findings, we propose that CNVs such as segmental duplications may generate more TADs, whose association with gene expression and functional consequences are worth further studying.” (page 11)

- “RPMI-8226 and U266 cells have different numbers of chromosomes, but similar number of TADs”. Isn't that expected, considering that TADs are called using ICE-normalized data that accounts for copy number changes?

Answer: We agree. We revised the sentences to (page 11):

“RPMI-8226 and U266 cells have different numbers of aneuploid chromosomes but similar numbers of TADs. Since the ICE method normalizes the Hi-C interactions to the per-copy level, CNVs may be responsible for the increased TAD numbers in MM cells compared to normal B cells.”

- The summary paragraph is too vague and unfocused.

Answer: Thanks for your suggestion. We have revised the summary paragraph to be more succinct (page 12).

11. In the Materials and Methods part of the paper, authors should include information about other datasets used in this study, e.g. GM 12787 Hi-C and RNA-seq and ChIP-seq data.

Answer: Thanks for pointing this out. We have added the following information in the “Materials and Methods” section (page 15):

The WGS dataset, RNA-seq dataset and Hi-C dataset of GM12878 were obtained by accession number ERX069505, GSM758560 and GSE63525 from the NCBI website. The ChIP-seq data of U266 were obtained from NCBI BioProject PRJEB1912/ERS333898. The ChIP-seq data of GM12878 were download from UCSC Table Browser. The manually curated CNV data were obtained from the CNVD database (<http://bioinfo.hrbmu.edu.cn/CNVD>).

We thank all the reviewers for their insightful suggestions that have helped improve this manuscript!

References

- 1 Dixon, J. R. *et al.* Topological domains in mammalian genomes identified by analysis of chromatin interactions. *Nature* **485**, 376-380, doi:10.1038/nature11082 (2012).

REVIEWERS' COMMENTS:

Reviewer #1 (Remarks to the Author):

Accept.

Reviewer #2 (Remarks to the Author):

All questions were addressed. I have no further questions.

Response to Reviewers

We thank the reviewers and editor for the critical comments, which have greatly helped us improve the results and manuscript. Please see below for our point-to point responses.

Reviewer #1 (Remarks to the Author):

Accept.

Reviewer #2 (Remarks to the Author):

All questions were addressed. I have no further questions.